# Plasma Metabolomic Profiling in 1391 Subjects with Overweight and Obesity from the SPHERE Study

**DOI:** 10.3390/metabo11040194

**Published:** 2021-03-24

**Authors:** Gianfranco Frigerio, Chiara Favero, Diego Savino, Rosa Mercadante, Benedetta Albetti, Laura Dioni, Luisella Vigna, Valentina Bollati, Angela Cecilia Pesatori, Silvia Fustinoni

**Affiliations:** 1EPIGET—Epidemiology, Epigenetics, and Toxicology Lab, Department of Clinical Sciences and Community Health, University of Milan, 20122 Milan, Italy; gianfranco.frigerio@unimi.it (G.F.); chiara.favero@unimi.it (C.F.); diego.savino@studenti.unimi.it (D.S.); rosa.mercadante@unimi.it (R.M.); benedetta.albetti@unimi.it (B.A.); laura.dioni@unimi.it (L.D.); valentina.bollati@unimi.it (V.B.); angela.pesatori@unimi.it (A.C.P.); 2Occupational Health Unit, Fondazione IRCCS Ca’ Granda Ospedale Maggiore Policlinico, 20122 Milan, Italy; luisella.vigna@policlinico.mi.it

**Keywords:** overweight, obesity, body mass index, metabolomics, plasma metabolome

## Abstract

Overweight and obesity have high prevalence worldwide and assessing the metabolomic profile is a useful approach to study their related metabolic processes. In this study, we assessed the metabolomic profile of 1391 subjects affected by overweight and obesity, enrolled in the frame of the SPHERE study, using a validated LC–MS/MS targeted metabolomic approach determining a total of 188 endogenous metabolites. Multivariable censored linear regression Tobit models, correcting for age, sex, and smoking habits, showed that 83 metabolites were significantly influenced by body mass index (BMI). Among compounds with the highest association, aromatic and branched chain amino acids (in particular tyrosine, valine, isoleucine, and phenylalanine) increased with the increment of BMI, while some glycerophospholipids decreased, in particular some lysophosphatidylcholines (as lysoPC a C18:2) and several acylalkylphosphatidylcholines (as PC ae C36:2, PC ae C34:3, PC ae C34:2, and PC ae C40:6). The results of this investigation show that several endogenous metabolites are influenced by BMI, confirming the evidence with the strength of a large number of subjects, highlighting differences among subjects with different classes of obesity and showing unreported associations between BMI and different phosphatidylcholines.

## 1. Introduction

Overweight and obesity are defined as an excessive fat accumulation able to impair human health. These problematic conditions have been almost tripled in the last decades and they are currently one of the most relevant global public heath burdens and major risk factors for noncommunicable disease such as type 2 diabetes, cardiovascular diseases, musculoskeletal disorders, and also some forms of cancer [1]. Body mass index (BMI) is calculated by dividing an individual’s weight (kg) by the square of their height (meters). Although it is not a perfect measure of a person’s fat accumulation, since it does not take into account the difference between body fat and lean body mass, it is easy to obtain and it is widely applied in both clinical evaluations and epidemiological studies to categorize an individual in one of the following groups: underweight (BMI below 18.5 kg/m^2^), normal weight (18.5–24.9), overweight (25.0–29.9), class I obesity (30.0–34.9), class II obesity (35.0–39.9), and class III obesity (above 40) [2].

To investigate the metabolic processes related to obesity, an approach could be the study of the metabolome, which is the ensemble of all the small molecules present in a biological fluid generated from the multitude of metabolic reactions of an organism [3]. Metabolomics is the comprehensive study of metabolites and is regarded as a promising approach to acquire a functional knowledge of the system’s biochemistry [4]. Recent advancements in analytical instrumentation allow to conduct the simultaneous determination of hundreds of metabolites in a short amount of time and requiring a small quantity of biological sample.

Previous studies, summarized by Rangel-Huerta et al. [5], have tried to outline the metabolomic signature of human obesity. In individuals with obesity elevated levels of branched-chain and aromatic amino acids and decreased levels of glutamine and glycine have been reported; also glucose is usually increased; while, considering lipids, alterations of glycerophospholipids, among which lysophosphatidylcholines lysoPC 18:0, 18:1, 18:2, and various phosphatidylcholines and sphingomyelins have been described, although with differences across studies [5]. Most of the reported studies investigated only a limited number of subjects [5]. Overall, further studies with elevated numbers of subjects are required to confirm if the metabolites reported to be altered in individuals with higher BMI represent a specific metabolic signature of this problematic condition.

The aim of this work was to characterize the plasma metabolome in subjects affected by overweight and obesity. A large number of individuals, including subjects with overweight and all different classes of obesity, was investigated. A validated targeted metabolomic assay, measuring 188 metabolites belonging to amino acids, biogenic amines, sum of hexoses, acylcarnitines, glycerophospholipids, and sphingolipids, was used.

## 2. Results

### 2.1. Study Population

The main demographic and clinical characteristics of the 1391 examined subjects are reported in Table 1. Age ranged from 18 to 86 years, with most of the participants aged from 30 to 69 years (84.5%). The population was predominantly composed of females (82%). The mean (±standard deviation) of BMI was 33.3 ± 5.5 kg/m^2^. In total, 397 (28.5%) subjects had a BMI less than 30 kg/m^2^, while 994 (71.5%) greater; among the latter, 530 (38.1%) had values of BMI ranging from 30.0 to 34.9 kg/m^2^ (class 1 obesity), 304 (21.9%) from 35.0 to 39.9 kg/m^2^ (class 2 obesity), and 160 (11.5%) greater than 40.0 kg/m^2^ (class 3 obesity). About half of subjects was never smokers (50.9%) and the majority of them was employed (60.4%). Further details about the population can be found in the paper describing the rationale and study protocol of the SPHERE study [6].

### 2.2. Metabolite Levels

The levels of the quantified metabolites in the study subjects are reported in Appendix A. Amino acids levels were quantifiable in all subjects, except for aspartic acid and citrulline, which showed a few concentrations below LOQ (3% and 0.1%). Among biogenic amines, creatinine, kynurenine, 4-hydroxyproline, and taurine were quantifiable in all subjects, while N-acetylornithine, asymmetric dimethylarginine, alpha-aminoadipic acid, L-dopa, methionine sulfoxide, putrescine, symmetric dimethylarginine, serotonin, spermidine, and total dimethylarginine were quantifiable in at least 20% of subjects. Carnitine (C0) and acetylcarnitine (C2) were quantifiable in all samples, while propionylcarnitine (C3), butyrylcarnitine (C4), valerylcarnitine (C5), hexanoylcarnitine (C6 (C4:1-DC)), octanoylcarnitine (C8), dodecanoylcarnitine (C12), dodecenoylcarnitine (C12:1), tetradecenoylcarnitine (C14:1), hexadecanoylcarnitine (C16), octadecenoylcarnitine (C18:1), and octadecadienylcarnitine (C18:2) were the acylcarnitines quantifiable in at least 20% of subjects. Among glycerophospholipids, only lysoPC a C14:0 and PC ae C42:0 were excluded from statistical analyses since they were above LOD in less than 20% of samples.

### 2.3. Impact of BMI on the Metabolomic Profile

The full results of the multivariable Tobit linear regression models are reported in Appendix A. Table 2 shows only the metabolites significantly associated with BMI. A graphical representation with a volcano plot is reported in Figure 1, which represents the % increase of each metabolite in relation to the increase of BMI. Volcano plots for the other independent variables (age, sex, and smoking habits) are reported in the Appendix A.

Several amino acids were positively associated with an increased BMI, in particular: tyrosine (percent of variation +29.8%, false discovery rate (FDR) *p* < 0.001), valine (+23.2%, FDR *p* < 0.001), isoleucine (+20.0%, FDR *p* < 0.001), phenylalanine (+20.4%, FDR *p* < 0.001), alanine (+18.9%, FDR *p* < 0.001), proline (+18.1%, FDR *p* < 0.001), glutamic acid (+17.1%, FDR *p* < 0.001), leucine (+15.0%, FDR *p* < 0.001), ornithine (+9.2%, FDR *p* = 0.002), lysine (+7.1%, FDR *p* = 0.026). Others were inversely associated with BMI: asparagine (−16.1%, FDR *p* < 0.001), glycine (−13.6%, FDR *p* < 0.001), histidine (−11.8%, FDR *p* < 0.001), serine (−10.0%, FDR *p* < 0.001), citrulline (−6.2%, FDR *p* = 0.030).

Among biogenic amines, kynurenine (+16.1%, FDR *p* < 0.001), aminoadipic acid (+18.5%, FDR *p* = 0.001), and 4-hydroxyproline (+8.0%, FDR *p* = 0.012) were positively associated with BMI, while serotonin (−7.4%, FDR *p* = 0.018), creatinine (−6.1%, FDR *p* = 0.023), and N-acetylornithine (−9.2%, FDR *p* = 0.030) were negatively associated with BMI.

The sum of hexose was positively associated with BMI (+17.8%, FDR *p* < 0.001).

Some acylcarnitines were positively associated with BMI: carnitine (C0), propionylcarnitine (C3), acetylcarnitine (C2), and valerylcarnitine (C5) (+12.6%, FDR *p* < 0.001; +12.2%, FDR *p* < 0.001; +9.1%, FDR *p* = 0.003; + 6.5%, FDR *p* = 0.037); dodecanoylcarnitine (C12), instead, was negatively associated with BMI (−24.5%, FDR *p* = 0.032).

Among lysophosphatidylcholines (lyso PC), lysoPC a C18:2, lysoPC a C18:1, and lysoPC a C17:0 were negatively associated with BMI (−23.3%, −15.9%, and −14.3%, FDR *p* < 0.001); while lysoPC a C16:1 was positively associated with BMI (+7.3%, FDR *p* = 0.021).

Only a few diacylphosphatidylcholines (PC aa) were positively associated with BMI: PC aa C38:3 (+21.1%, FDR *p* < 0.001), PC aa C40:4 (+14.3%, FDR *p* < 0.001), PC aa C32:1 (+12.4%, FDR *p* < 0.001), PC aa C38:4 (+7.6, FDR *p* = 0.015), and PC aa C40:5 (+6.9%, FDR *p* = 0.028); while a total of 12 diacylphosphatidylcholines were negatively associated, among which PC aa C38:6 (−12.9, FDR *p* < 0.001) and PC aa C38:0 (−11.7%, FDR *p* < 0.001) were the highest.

No acylalkylphosphatidylcholine (PC ae) was positively associated with BMI, while 27 were negatively associated; in particular, PC ae C36:2 (−20.5%, FDR *p* < 0.001), PC ae C34:3 (−20.2%, FDR *p* < 0.001), PC ae C34:2 (−18.2%, FDR *p* < 0.001), and PC ae C40:6 (−16.1%, FDR *p* < 0.001) were those with the highest variation.

Finally, only two sphingomyelins (SM) were positively associated with BMI: SM C18:1 (+8.3%, FDR *p* = 0.006) and SM C16:1 (+7.0%, FDR *p* = 0.019), while six were negatively associated: SM C24:1 (−10.5%, FDR *p* < 0.001), SM C16:0 (−9.7%, FDR *p* = 0.001), SM C26:1 (−8.9%, FDR *p* = 0.002), SM (OH) C22:2 (−8.3%, FDR = 0.002), SM (OH) C16:1 (−6.7%, FDR *p* = 0.021), and SM (OH) C14:1 (−5.9%, FDR *p* = 0.041).

### 2.4. Metabolite Distributions and Correlations

Boxplots of the 16 most significant metabolites stratified by BMI are reported in Figure 2. Most metabolites showed a linear trend; tyrosine (tyr), valine (val), isoleucine (ile), PC aa 38:3, phenylalanine (phe), alanine (ala), and the sum of hexose (H1) showed increased concentrations among groups with increased BMI, while lysoPC a C18:2, PC ae C36:2, PC ae C34:3, PC ae C34:2, PC ae C40:6, asparagine (asn), PC ae C40:1, lysoPC a C18:1, and PC aa C38:0 decreased. Considering all the metabolites included in the statistical analyses, a principal component analysis showed no clear separation among subjects grouped by BMI classes (Appendix A).

The network analyses are reported in Figure 3 and Figure 4. In Figure 3 only those metabolites that correlated one to each other with a Pearson’s r > 0.4 are reported. In general, we note that metabolites belonging to the same class are grouped together. Most amino acids were intercorrelated; in particular phenylalanine, alanine, tyrosine, methionine, and tryptophan showed a good correlation (Pearson’s r > 0.6), while a higher correlation (Pearson’s r > 0.8) was found among leucine, isoleucine and valine. All lipid classes were grouped together, with the exception of acylcarnitines which were closer to amino acids, in particular C3 with valine (r = 0.43) and isoleucine (r = 0.47). The hexose also had a good correlation with isoleucine (r = 0.41) and leucine (r = 0.41). Among biogenic amines, alpha-amino adipic acid correlated with isoleucine (r = 0.42), kynurenine was related to tyrosine (r = 0.41), while serotonin was correlated with taurine (r = 0.57), which was correlated with spermidine (r = 0.46) and with glutamic acid (r = 0.47), in turn related to aspartic acid (r = 0.43). Dimethylarginine (total DMA) showed a good correlation with SM C26:0 (r = 0.44). In Figure 4 metabolites that are correlated one to each other with a Pearson’s r > 0.7 are reported. Among amino acids, only leucine, isoleucine and valine are included. Several lysoPCs were correlated each other as well as several SMs. Among PCs, those with similar number of carbons were more closely related. Similar results can be observed in the Appendix A with the cluster analysis (dendrogram) (Appendix A).

A heat map considering only the metabolites most significantly associated with BMI in the Tobit models is reported in Figure 5. The dendrogram reported is obtained with a cluster analysis and shows how these metabolites are related to each other. Metabolites grouped on the right part of the picture (including the hexose, kynurenine, C0, C3, and some amino acids) showed higher levels with the increase of BMI of subjects, vice versa those grouped on the left part, as lyso PCs, glycine and most PC, showed lower level with the increase of BMI.

### 2.5. Possible Involved Biochemical Pathways

Figure 6 reports a visual representation of the main metabolic pathways which could be significantly altered among individuals with different BMI, according with the SMPDB library. The complete table resulting from the pathway analysis is reported in the Appendix A. Some metabolic pathways involved in the metabolism of amino acids might be altered among individuals with different BMI, including valine, leucine, and isoleucine degradation; phenylalanine and tyrosine metabolism; aspartate metabolism. Further pathways altered might be oxidation of branched and very long chain fatty acids, ammonia recycling, carnitine synthesis, catecholamine biosynthesis, bile acid biosynthesis, and porphyrin metabolism.

## 3. Discussion

In this work we described the application of a targeted metabolomic approach to a large cohort of subjects affected by overweight and obesity with the aim of describing their metabolomic profile. The main result of this study is the identification of several differences in terms of metabolite concentrations among subjects with different BMI.

Among metabolites significantly associated with BMI, the branched chain amino acids (BCAAs) valine, isoleucine, and leucine were positively associated. BCAAs have been reported to be associated with BMI in several studies [7,8,9,10,11,12,13,14,15,16]. The results of our study further confirm that an increased BMI is associated with increased levels of these compounds also among subjects with overweight and obesity. BCAAs has been suggested to be strongly connected with well-known consequences of obesity: insulin resistance and diabetes [17,18]. Some reasons for the increase of circulating BCAA have been postulated: one is the suppression of the enzymatic catabolism of BCAAs in the adipose tissue and liver of individuals with obesity [18,19] as, in particular, the insulin-induced impairment of branched-chain α-keto acid dehydrogenase (BCKD) [20]. Others have suggested that high circulating levels of BCAAs might be one of the causes of insulin resistance through the activation of the mammalian target of rapamycin (mTOR) signaling [17,21,22].

The aromatic amino acids tyrosine and phenylalanine were also strongly related with BMI; the first, in particular, was the most significant metabolite in our analysis being associated with BMI (FDR *p*-value < 0.0001, +29.8% of variation). Higher levels of aromatic amino acids in individual with higher BMI have been found in other studies [5,7,11,12,13,16]. Tyrosine can originate from phenylalanine [23]. Similarly to what has been proposed for branched chain amino acids, also the mechanism proposed for the increased level of tyrosine is the BCKD inhibition [20].

We did not find higher concentrations of tryptophan in subjects with increased BMI, as opposed by some other studies [8,16,24], instead we observed an association between BMI and kynurenine, which is a metabolite of tryptophan [5], as also observed by other studies [13,25,26]. Indeed, alteration in kynurenine pathway has been related with BMI, insulin resistance, and with the low-grade systemic inflammation characteristic of obesity [5,27].

In addition, the hydrophobic amino acids alanine and proline were positively associated with BMI. Similar observations were reported in previous studies (for alanine [7,10,11,13]; for proline [7,10]). Other amino acids increased with BMI were glutamic acid (in agreement with other studies [9,10,16,25,26]) and lysine (as observed in previous reports [7,11]). Maltais-Payette and coworkers, in particular, found that glutamate is strongly associated with visceral adipose tissue [28]. Furthermore, the nonproteinogenic amino acid ornithine was higher and this association was observed also by Dunn et al. [7].

Amino acids negatively associated with BMI were the polar amino acids asparagine, glycine, histidine, serine, and citrulline. Some other studies also reported some of these negative associations [7,10], while these results were in disagreement with others: Oberbach et al. found a significant positive association with glycine [29].

Higher levels of the sum of hexose were found in individuals with higher BMI; this is not surprising since the main hexose in human blood is glucose, and glucose impairment related to metabolic syndrome and type 2 diabetes is one of the main consequences of obesity [30].

Considering biogenic amines, in addition to kynurenine, already discussed above, according with the Tobit regression models, aminoadipic acid and 4-hydroxyproline were positively associated while serotonin and creatinine were negatively associated to BMI. However levels of creatinine are not in agreement with Dunn et al., where serum creatinine levels were positively associated with BMI [7].

Some acylcarnitines were significantly higher in individuals with higher BMI: C0, C3, C2, and C5, some of which in agreement with other studies [11,12,14]. An incomplete beta-oxidation of fatty acids may be the cause of the increased levels of acylcarnitines [5,18,31,32]. Furthermore, C3 and C5 are by-products and intermediates of the BCAAs isoleucine and leucine catabolism [5,33]. Indeed, C3 was strongly correlated with leucine and isoleucine.

LysoPC 16:1 was the only lysoPC positively associated with BMI (in agreement with Bagheri and coworkers [10]). LysoPC C18:2 and lysoPC C18:1, were negatively correlated with BMI, in agreement with several other studies [8,9,10,13,16,25,34]. LysoPC a C18:2, in particular, was the second metabolite (after tyrosine) with the lowest FDR *p*-value (<0.0001, −23.3% of percent variation) in the Tobit model. Due to this well-established evidence, lysoPC a C18:2 could be considered one of the most interesting negative biomarkers of obesity.

Only a few PC aa increased with BMI (PC aa C38:3, PC aa C40:4, PC aa C32:1, PC aa C38:4, and PC aa C40:5) while many decreased. In agreement with our results, Ho et al. showed positive association with PC aa C38:3 and negative association with PC aa 38:6 [13]. Our results agreed with Bagheri and coworkers in finding higher levels of PC aa C32:1 and PC aa C38:3 in individuals with obesity [10]. Carayol et al., assessing the metabolomic profile in two cohorts, were in agreement with our results in finding a positive association with PC aa C32:1, PC aa C38:3, PC aa C38:4 and negative association with PC aa C42:0, PC aa C42:1, and PC aa C42:2 [25]. Conversely, Oberach et al. found higher level of PC aa 42:0 in subjects with obesity, while we found a significantly inverse association of this compound with BMI, furthermore they found PC aa 32:1 and PC aa 40:5 decreased in subjects with obesity while we found a positive association of these metabolites with BMI [29].

No PC ae increased with BMI while many decreased. Those in common with Bagheri were PC ae C34:3, PC ae C38:4 e PC ae C40:6 [10]. Carayol et al. found negative association between BMI and PC ae C38:2, PC ae C40:5, PC ae C42:3, PC ae C42:4, PC ae C42:5, PC ae C44:5, PC ae C44:6, and PC ae C44:4: those results were in agreement with ours except for the latter [25].

A possible reason for the decreased levels of lysoPCs and PCs in individuals with higher BMI might be related to an increase of their degradation. Indeed, Müller and coworkers reported that glycosylphosphatidylinositol-anchored proteins (GPI-AP) are released into circulation in rats and humans suffering from obesity and diabetes. As a protective mechanism to balance the deleterious effect of circulating GPI-AP, a serum glycosylphosphatidylinositol-specific phospholipase D is upregulated to degrade those proteins, thus reducing their circulating levels [35,36]. Since lysoPCs and PCs are associated with GPI-AP in micelle-like complexes, a similar increased lipolytic degradation could be postulated as an explanation of their reduced concentrations in individuals with higher BMI.

Among sphingomyelins, SM C18:1 and SM C16:1 were significantly associated with BMI. Ho et al. found positive association between BMI and these two compounds, finding also an association with SM 18:0, which was not significant in our population [13]; also Carayol et al. found a positive association with SM C18:0 [25]. Sphingomyelins significantly lower were SM C24:1, SM C16:0, SM C26:1, SM (OH) C22:2, SM (OH) C16:1, and SM (OH) C14:1. However, even though the evidence from this study is noteworthy considering the number of subjects considered, there is not a clear agreement across studies for most of the considered lipids, so most of them cannot be considered critical biomarkers of obesity, unless further validation.

Trabado and coworkers performed the same targeted metabolomic analysis in 800 healthy subjects in order to obtain reference values for the considered metabolites [37]. Since our population had higher BMI levels (interquartile rage of our population was 29.5–36.3 kg/m^2^, while in the healthy population was 21.7–25.4 for males and 20.4–24.0 kg/m^2^ for females, respectively), we compared our results with the proposed reference values to observe differences in the metabolome between populations with such a different distribution of BMI (Appendix A). Some amino acids showed higher levels in our population than the proposed reference values: glutamic acid, ornithine, phenylalanine, tyrosine, leucine, lysine, valine, and alanine. These amino acids, indeed, were significantly higher in individuals with higher BMI in our Tobit models. Asparagine and arginine were considerably lower in our population than reference values; only asparagine was also negatively associated with BMI in our Tobit model, while the results for arginine were not significant. LysoPC a C26:0 and lysoPC a C28:0 were considerably higher in reference values than our population, while we found no significant differences. Interestingly, though, Carayol et al. did find a positive association between BMI and lysoPC a C28:0 [25]. It is noteworthy that most phosphatidylcholines, especially PC ae, were considerably lower than reference values. SM C18:0 and SM C18:1 were higher in our population than the reference values; among those SM C18:1 was actually significantly associated with BMI in the Tobit models while SM C18:0 was not, while Ho et al. and Carayol et al. did find a positive association for this metabolite [13,25]. SM C26:0 and SM C26:1 were considerably lower in this population than reference values, but only the latter was also significantly correlated with BMI in the Tobit model. However, a limitation should be considered when comparing our results with this reference values since sex was another variable consistently different among the two populations (82.1% of females in our population vs. 47.9% of females in the study of healthy volunteers). Finally, an overview of results and a comparison with the literature is reported in Appendix A.

Results of the cluster analysis performed on metabolites significantly associated with BMI, reported in the heat map, showed that a cluster composed by branched chain and aromatic amino acids, kynurenine, sum of hexose (H1), carnitine (C0), and propionylcarnitine (C3), PC aa 40:4, 38:3, 32:1, and 42:5 have higher metabolite concentrations in individuals with higher BMI; while the other cluster of metabolites (containing glycine, many PC ae, and lysoPC,) tends to have lower metabolite levels with subjects with higher BMI, consistently with the other statistical analyses performed. Interestingly, histidine and asparagine, while being negatively associated with BMI in the linear regressions, were part of the main cluster of positively associated metabolites. However, the heat map does not show a clear separation of colors and this indicates that the complexity of the metabolome, which is determined by a multitude of factors, cannot be related exclusively to the BMI. This can also be observed with the result obtained with the PCA visualization.

The pathway analysis reported suggests which metabolic pathways might be altered in individuals with different BMI. The main altered pathways are related to amino acid metabolic pathways, in particular to aromatic and branched chain amino acids; others are related to oxidation of fatty acids (due to the alteration of acylcarnitines) and, indeed, a defective fatty acid oxidation in subjects with obesity has been reported [38]. Other altered pathways evidenced are ammonia recycling (due to alterations of glycine, aspartic acid, asparagine, serine, and histidine), catecholamine biosynthesis (due to alterations of tyrosine), carnitine synthesis (due to alterations in carnitine, glycine, and lysine), porphyrin metabolism (due to alterations of glycine and alanine), bile acid biosynthesis (due to alterations of glycine, alanine, and taurine); nevertheless, the metabolites responsible for the supposed pathway alterations are only a few compared with the total number of compounds in each pathway and the impact is low (0.21 or less) [39]. It is worth mentioning that this tool has some limitations, as it did not identify any altered pathways concerning the several altered glycerophospholipids, maybe because this information was not present in the pathway library used. Moreover, it was adopted only to have an insight of the possible altered metabolic pathways, which require adequate confirmation with dedicated studies.

One of the main strengths of this study is the high number of subjects included (more than a thousand), which, not only have allowed to find several significant associations with BMI, but could have also helped to reduce the entities of underlined bias affecting the metabolome. Moreover, the implemented metabolomic analyses allowed a robust absolute quantification and a good interlaboratory reproducibility [40], therefore the results are suitable for comparisons among studies, as we did with the reference values proposed by Trabado and coworkers [37].

This study has some limitations: the metabolome is highly influenced by several factors and we took in consideration only a few: indeed, some of the considered metabolites might be influenced by diet. This bias was only partially mitigated with the fasting collection of blood. Moreover, most of the subjects were female (82%) and important metabolomic differences have been reported among subjects with different sex: higher levels of SMs and PCs in females and higher levels of creatinine and branched-chain amino acids in males [37,41], also confirmed by our results (Appendix A); however, the Tobit linear regression models were corrected also for sex. Another limitation is the use of a targeted approach: even though a great number of metabolites was considered (more than 180), they do not cover the entire metabolome and other approaches, like untargeted metabolomics, could have evidenced alteration in other metabolites. Finally, the study aimed only to assess an association of the considered metabolites with the BMI of subjects, but other variables should be considered (as associations with metabolic complications of obesity). Furthermore, longitudinal analyses should be performed in order to find metabolites able to predict the onset of the negative consequences related to obesity. 

In conclusion, the present study assessed the metabolomic profile using a validated targeted metabolomic approach on the SPHERE cohort, composed by subjects affected by overweight and obesity. The results obtained evidenced several biomarkers related to obesity, most of which are a further confirmation of the body of evidence already present (as for BCAAs, tyrosine, lyso PC a 18:1 and 18:2), while several others have been firstly evidenced, in particular metabolites belonging to the classes of phosphatidylcholines and sphingomyelins.

## 4. Materials and Methods

### 4.1. Study Subjects

The subjects were enrolled in the frame of the SPHERE (“Susceptibility to Particle Health Effects, miRNAs and Exosomes”) project. Among the subjects of this cohort, 1391 had suitable data and plasma samples collected and were therefore included in this work. The study design and enrolment criteria were described previously [6]. Briefly, the study participants were recruited at the Center for Obesity and Work, Fondazione IRCCS Ca’ Granda Ospedale Maggiore Policlinico, Milan, Italy from 2010 to 2015. The eligibility criteria were (1) age: 18 years or older; (2) BMI: equal or greater than 25 kg/m^2^; (3) resident in Lombardy at the enrolment; (4) agreement to sign an informed consent and donate blood and urine samples. Exclusion criteria included previous diagnosis of cancer, heart disease, stroke, or other chronic diseases in the last year (such as multiple sclerosis, Alzheimer’s disease, Parkinson’s disease, depression, bipolar disorder, schizophrenia, and epilepsy). Weight and height were measured by a nurse following standardized procedures as part of the routine protocol. Biochemical parameters were also collected such as glycated haemoglobin. A questionnaire collected sociodemographic and lifestyle information, such as smoking status and medications use. The study was approved by the local Institutional Review Board (Fondazione IRCCS Ca’ Granda Ospedale Maggiore Policlinico review board). To reduce bias associated with obesity, we implemented people-first language according to the standard recommendation of European Association for the Study of Obesity (EASO), The Obesity Society (TOS) and Obesity Canada (OC) [42,43,44].

### 4.2. Plasma Sample Collection

Blood was collected into EDTA tubes on the morning of recruitment (between 8 and 10 a.m.) and transported to the laboratory within 2 h of phlebotomy; about 7.5 mL of blood was centrifugated at 1300× *g* for 15 min at room temperature to obtain the plasma. The supernatant was stored in aliquots at −80 °C until use.

### 4.3. Metabolomic Analyses

The analysis of the subjects’ metabolomic profile was conducted using an LC–MS/MS targeted metabolomic method using the AbsoluteIDQ^®^ p180 kit (Biocrates Life Sciences AG, Innsbruck, Austria) [45]. This tool allows a standardized assay for the determination of a total of 188 metabolites in plasma: 21 amino acids, 21 biogenic amines, the sum of hexose (H1), 40 acylcarnitines, 15 sphingolipids, and 90 glycerophospholipids. Among the latter, 14 are lysophosphatidylcholines (LysoPC), 38 are diacylphosphatidylcholine (PC aa), and 38 are acylalkylphosphatidylcholine (PC ae). The complete list of analyzed metabolites and the related abbreviations used in this article is reported in Appendix A. The good interlaboratory reproducibility of this assay for the measurements of these metabolites in human plasma has been reported [40].

The instrumentation consisted of a high-pressure liquid chromatograph Agilent 1260 (Agilent Technologies, Cernusco Sul Naviglio, Italy) coupled with a hybrid triple quadrupole/linear ion trap mass spectrometer (QTRAP 5500; AB Sciex, Monza, Italy) with an electrospray ionization source. The assay was conducted following the indications contained in the AbsoluteIDQ^®^ p180 kit manual and the instructions of the Biocrates application support specialists. Briefly, the samples were loaded on 96-well plates already containing the isotopic labeled internal standards of lipids, along with the blank samples (phosphate buffer solution), a 7-points calibration curve, and three levels of plasma quality control samples (the medium level was repeated at least three times in each plate). From 78 to 80 plasma samples were loaded in each plate. Before adding the samples, each well of the plate was added with the solution containing the isotopically labeled internal standards of amino acids and biogenic amines, dried, added with 50 µL of a phenyl isothiocyanate solution (solubilized in water, ethanol, and pyridine) for derivatization of amino acids and biogenic amines, incubated at room temperature for 20 min, dried, added with 300 µL of 5 mL ammonium acetate in methanol as extraction solvent. The plate was then agitated and centrifuged at 500× *g* for 2 min. From each well, an aliquot was taken, transferred, and diluted in the analysis plate used for the LC–MS/MS method, while another aliquot was transferred and diluted in the plate for FIA–MS/MS analysis. The LC–MS/MS analysis was used for the quantitation of amino acids and biogenic amines, it consisted of a linear gradient of water (A phase) and acetonitrile (B phase) containing formic acid, using an Agilent Zorbax Eclipse XDB C18 (3.0 × 100 mm, 3.5 µm) (Agilent Technologies, Santa Clara, CA, USA) equipped with a guard-column SecurityGuard, C18 (4 × 3 mm) (Phenomenex, CA, USA), while the mass spectrometry operated in scheduled multiple reaction monitoring (s-MRM), in positive polarity. The Analyst^®^ software (version 1.6.3; Ab Sciex S.r.l, Milano, Italy) was used to prepare the sequence of analyses, the MultiQuant^TM^ software (version 3.0.8664.0; Ab Sciex S.r.l, Milano, Italy) was used for the integration of chromatographic peaks; results were then elaborated with interpolation with the 7-points calibration curve and imported into the MetIDQ software (version 7.13.11-DB109-Nitrogen-2850; Biocrates Life Sciences AG, Innsbruck, Austria). All the lipid classes and the sum of hexose were analyzed within the FIA–MS/MS method. All the samples were directly injected in the mass spectrometer, which operated in multiple reaction monitoring, positive polarity, and quantified with a one-point calibration. The Analyst software was used for the preparation of analyses sequence while results were directly imported and elaborated with the MetIDQ software. For both methods (LC–MS/MS and FIA–MS/MS), the injection order was randomized. Precision and accuracy of the method were checked for each batch of analysis using the MetIDQ software. Limit of detections (LOD) were automatically calculated by MetIDQ for each plate through comparison with the blank sample. Even metabolite levels greater than the LOD but lower than the LLOQ (lower limit of quantification) or greater than the ULOQ (upper limit of quantitation) were calculated using the calibration curve in order to reduce the entity of missing data.

### 4.4. Statistical Analyses

Metabolomic data were batch-normalized using the MetIDQ software, with median values used for calculation of normalization factors. For each metabolite, data <LOD were replaced with the minimum LOD values among all plates; then, descriptive statistics were performed including mean  ±  standard deviation (SD), median, interquartile range, and extreme values. Only metabolites with at least 20% of observations greater than the LOD were considered for the following statistical analyses. The few missing values in metabolite concentrations were imputed using the k-nearest neighbors algorithm (k-NN) [46] with a value of k equal to 37 (about the square root of the total number of observations). Metabolite concentrations were log-transformed (base e) to ensure normal distribution and then standardized performing an auto-scale (each value was subtracted by the mean and divided by the standard deviation) to make the effects of BMI on the different metabolites comparable.

We used multivariable censored linear regression models (Tobit) to test the relationship between metabolite concentrations and subject characteristics. The Tobit model, a censored regression model, is designed to estimate linear relationships between variables when there is left- or right-censoring in the dependent variable [47,48]. In our case, metabolite concentrations lower than the limit of detections (LOD) were considered as the left-censored values. A Tobit linear regression model was estimated for each metabolite: the standardized natural logarithm of the metabolite concentrations was the dependent variable, while independent variables were age, sex (male = reference, or female), body mass index (BMI) (kg/m^2^), and smoking status (non-smokers = reference, former smokers, or current smokers). We calculated standardized beta coefficients to estimate and compare the individual and independent effects of all predictors. The percent of variation (∆%) was also calculated through the formula: (exp(β) − 1) × 100, where β was the regression coefficient representing the increase of the metabolite in relation to the one unit increase of the independent variable. The *p*-values were then adjusted for multiple testing by controlling the false discovery rate (FDR) according to the method of Benjamini and Hochberg [49]. A two-sided *p*-value of 0.05 was considered as statistically significant. These statistical analyses were performed with SAS software (version 9.4; SAS Institute Inc., Cary, NC, USA).

Data visualization was performed using R (R version 4.0.2, R Foundation, Vienna, Austria) [50] with the Rstudio interface (Version 1.3.959, RStudio Inc., Boston, MA, USA) and the package “tidyverse” [51]. Volcano plots of Δ% vs. −log_10_(FDR *p*-values) were used to display results of the Tobit analyses. Boxplots were built to visualize how the distribution of metabolite concentrations differs among different BMI classes, considering the 16 analytes with the lowest FDR *p*-values for BMI in the Tobit models. Network analyses were performed to visualize how the metabolites correlate each other: the packages “tidygraph” and “ggraph” [52,53] were implemented; metabolites were considered as nodes and correlation coefficients (r) obtained from each pair of metabolites as edges; the Fruchterman–Reingold force-directed layout algorithm was used and edge weights were set on the value of r; only statistically significant correlations with r > 0.4 and 0.7 were considered and metabolites with no connection were removed: these cut-offs were chosen to have an optimal overview of most of the considered metabolites (r > 0.4) and to have a better separation and visualization of glycerophospholipids and sphingolipids (r > 0.7). A principal component analysis (PCA) was performed with “ggfortify” package [54] to visualize if overall metabolites variation was related to the different classes of BMI. Implementing the “ape” package [55], cluster analysis was used for building dendrograms with Euclidean distances among metabolites. A heat map was built considering only the most significant metabolites (FDR < 0.0001) in the Tobit regressions, which were grouped by a cluster analysis, and the subjects, sorted by BMI.

Finally, a pathway analysis was performed in order to have an insight of the altered metabolic pathways among subjects using the web-based tool MetaboAnalyst [39]. An HMDB code [56] was assigned to each metabolite as long as it was available, a regression was performed to compare data with BMI of subjects, a GlobalTest was selected as pathway enrichment analysis, an out-degree centrality was chosen as pathway topology analysis, and the small molecule pathway database (SMPDB) Homo Sapiens library [57] was chosen as pathway library.

## Figures and Tables

**Figure 1 metabolites-11-00194-f001:**
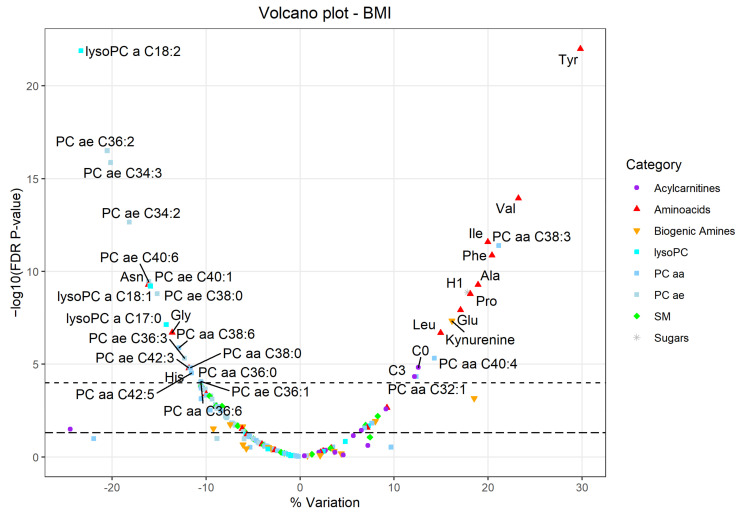
Volcano plot representing the results of the Tobit linear regression models considering the metabolites (dependent variables) in relation to BMI (independent variable), adjusted for age, sex, and smoking habit. Each dot represents a metabolite and they are displayed based on the % variation (∆% = (exp(β) − 1) × 100) (*x*-axis) and the negative logarithm (base 10) of the FDR *p*-value (*y*-axis). The upper dashed line represents an FDR *p*-value equal to 0.0001, while the lower dashed line represents an FDR *p*-value equal to 0.05.

**Figure 2 metabolites-11-00194-f002:**
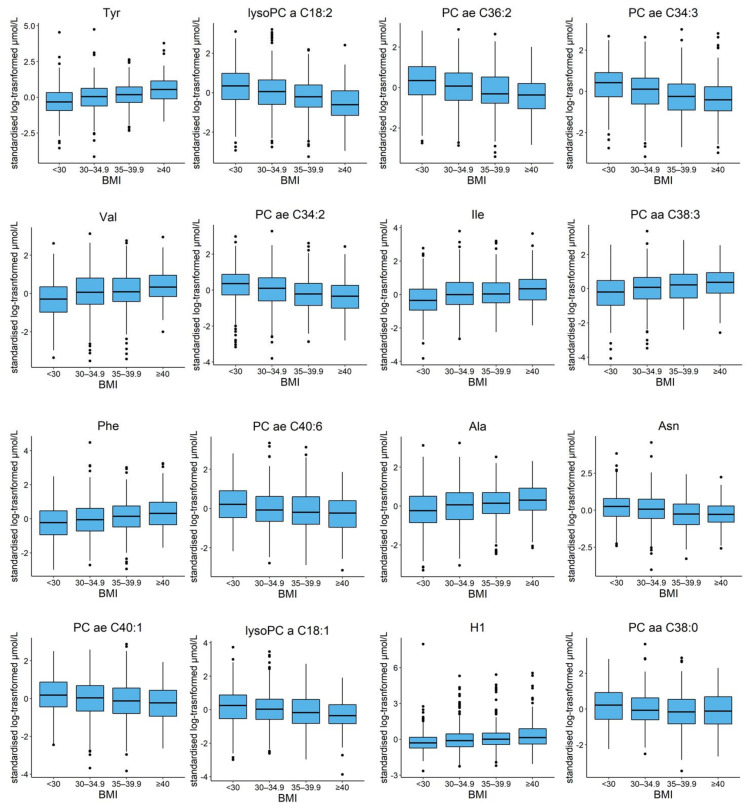
Boxplots summarizing the distribution, for study subjects divided in four different classes of BMI, of the 16 metabolites with the lowest FDR *p*-value in the Tobit regression models. The box contains 50% of the observations, with the median dividing the box in two areas and the upper and lower hinge representing the 25th and 75th percentile of the distribution. Outside the box, the upper whisker extends from the hinge to the highest value no further than 1.5 times the interquartile range (IQR) from the hinge. The lower whisker extends from the hinge to the smallest value at most 1.5 times the IQR of the hinge. Data beyond the whiskers are plotted individually and represented as dots.

**Figure 3 metabolites-11-00194-f003:**
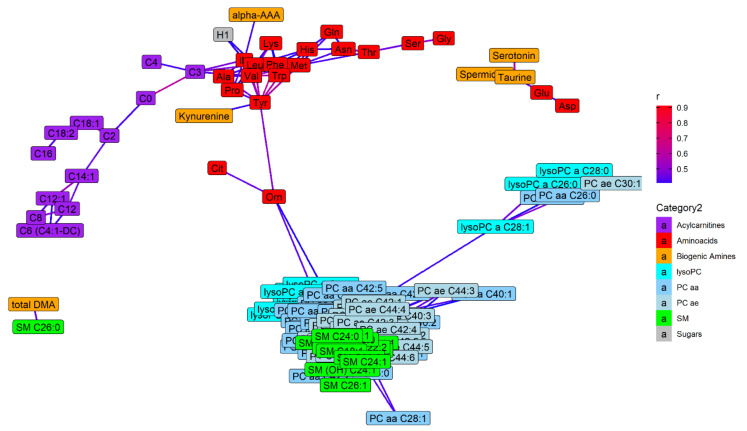
Network analysis performed considering metabolites as nodes and correlation coefficients (r) obtained from each pair of metabolites as edges. The Fruchterman–Reingold force-directed layout algorithm was used, and the edge weights were set on the value of r. Only statistically significant correlations with r > 0.4 were considered and metabolites with no connection were removed.

**Figure 4 metabolites-11-00194-f004:**
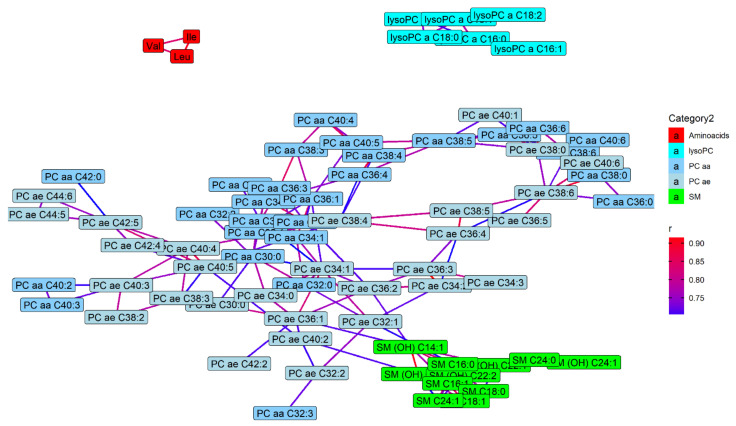
Network analysis performed considering metabolites as nodes and correlation coefficients (r) obtained from each pair of metabolites as edges. The Fruchterman–Reingold force-directed layout algorithm was used, and the edge weights were set on the value of r. Only statistically significant correlations with r > 0.7 were considered and metabolites with no connection were removed.

**Figure 5 metabolites-11-00194-f005:**
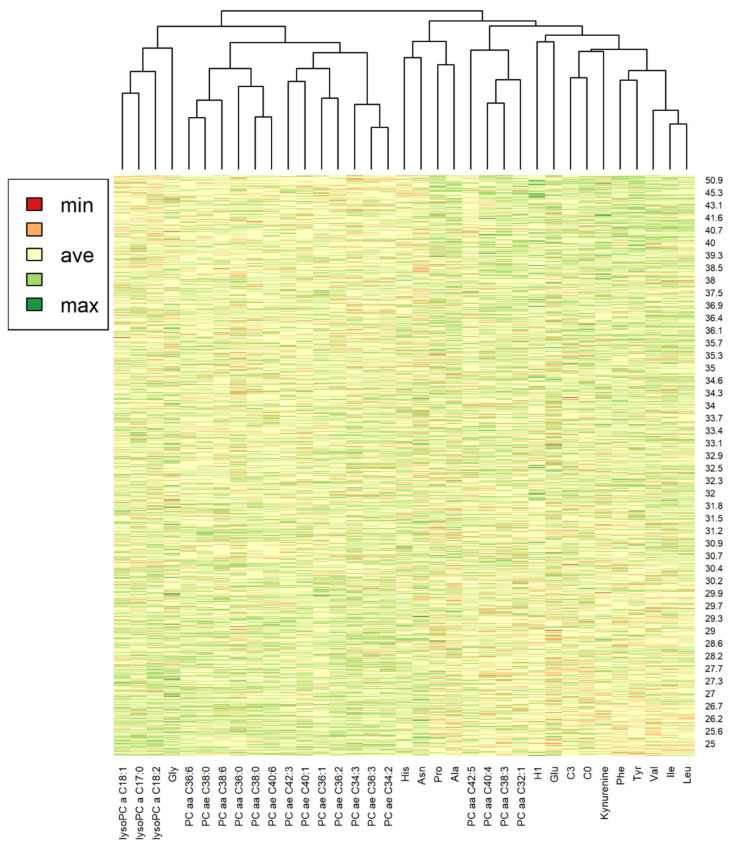
Heat map showing metabolite levels among subjects. The subjects were sorted by BMI. Only the most significant metabolites in the Tobit models for BMI (FDR *p*-value < 0.0001) were considered and they were grouped with a cluster analysis. Dendrograms built with Euclidean distances related to the cluster analysis are reported above.

**Figure 6 metabolites-11-00194-f006:**
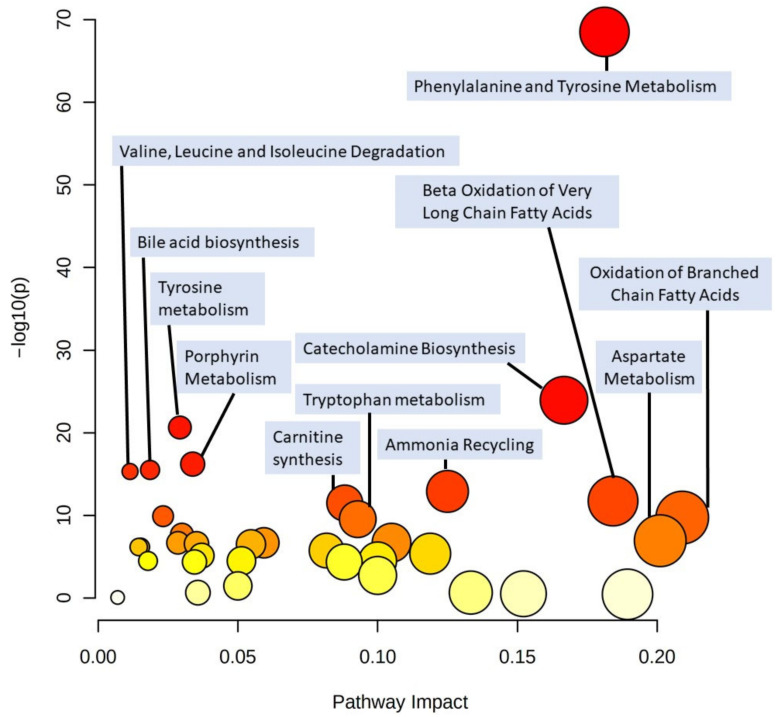
Plot relative to the pathway analysis displaying each altered pathway as a dot, ordered for pathway impact (*x*-axis and size) and negative logarithm (base 10) of the *p*-value (*y*-axis and color). The pathway analysis was performed with regressions between metabolites and the BMI of subjects, a GlobalTest was selected as pathway enrichment analysis, an out-degree centrality was chosen as pathway topology analysis, and the SMPDB Homo sapiens library was chosen as pathway library.

**Table 1 metabolites-11-00194-t001:** Demographic and personal characteristics of the studied population.

		All	BMI <30	BMI 30–34.9	BMI 35–39.9	BMI ≥40
*N*		1391	397	530	304	160
Age, years (mean ± SD)		51.8 ± 13.5	50.3 ± 13.6	52.3 ± 13.6	52.8 ± 13.6	52.3 ± 12.6
Ages, category	18–29	101 (7.3%)	39 (9.8%)	34 (6.4%)	18 (5.9%)	10 (6.2%)
	30–49	486 (34.9%)	150 (37.8%)	180 (34.0%)	106 (34.9%)	50 (31.2%)
	50–69	689 (49.5%)	187 (47.1%)	260 (49.1%)	150 (49.3%)	92 (57.5%)
	70–89	115 (8.3%)	21 (5.3%)	56 (10.6%)	30 (9.9%)	8 (5.0%)
Gender	Males	250 (18.0%)	52 (13.1%)	112 (21.1%)	61 (20.1%)	25 (15.6%)
	Females	1141 (82.0%)	345 (86.9%)	418 (78.9%)	243 (79.9%)	135 (84.4%)
Smoking status	Never smoker	705 (50.7%)	207 (52.1%)	259 (48.9%)	155 (51.0%)	84 (52.5%)
	Former smoker	476 (34.2%)	130 (32.8%)	182 (34.3%)	108 (35.5%)	56 (35.0%)
	Current smoker	201 (14.4%)	57 (14.3%)	85 (16.0%)	41 (13.5%)	18 (11.3%)
	N.A.	9 (0.7%)	3 (0.8%)	4 (0.8%)	-	2 (1.2%)
Occupation	Employee	830 (59.7%)	244 (62.5%)	329 (62.1%)	177 (58.2%)	80 (50.0%)
	Unemployed	125 (9.0%)	39 (9.8%)	40 (7.5%)	24 (7.9%)	22 (13.8%)
	Pensioner	309 (22.2%)	81 (20.4%)	118 (22.3%)	75 (24.7%)	35 (21.9%)
	Homemaker	111 (8.0%)	29 (7.3%)	38 (7.2%)	22 (7.2%)	22 (13.8%)
	N.A.	16 (1.1%)	4 (1.0%)	5 (0.9%)	6 (2.0%)	1 (0.6%)

**Table 2 metabolites-11-00194-t002:** Metabolites significantly associated with body mass index (BMI) in the censored linear regression Tobit models. The dependent variable was the log transformed and standardized concentration of a given metabolite, while independent variables were BMI, age, sex, and smoking habit. For each metabolite, the percent of variation was calculated with the formula: (exp(β) − 1) × 100, where β was the slope representing the increase of the metabolite in relation to the increase of BMI. The *p*-values adjusted for multiple testing by controlling the false discovery rate (FDR) are also reported. Only metabolites with FDR *p*-values lower than 0.05 were included in this table; complete results are reported in Appendix A.

**Positively Associated**			
**Category**	**Metabolite**	**% Variation**	**FDR *p*-Value**
Aminoacids	Tyrosine (Tyr)	29.8	1.01 × 10^−22^
Aminoacids	Valine (Val)	23.2	1.12 × 10^−14^
Aminoacids	Isoleucine (Ile)	20.0	2.59 × 10^−12^
PC aa	PC aa C38:3	21.1	4.02 × 10^−12^
Aminoacids	Phenylalanine (Phe)	20.4	1.35 × 10^−11^
Aminoacids	Alanine (Ala)	18.9	5.14 × 10^−10^
Sugars	Sum of hexose (H1)	17.8	1.31 × 10^−9^
Aminoacids	Proline (Pro)	18.1	1.62 × 10^−9^
Aminoacids	Glutamic acid (Glu)	17.1	1.18 × 10^−8^
Biogenic Amines	Kynurenine	16.1	4.64 × 10^−8^
Aminoacids	Leucine (Leu)	15.0	2.03 × 10^−7^
PC aa	PC aa C40:4	14.3	4.71 × 10^−6^
Acylcarnitines	Carnitine (C0)	12.6	1.5 × 10^−5^
Acylcarnitines	Propionylcarnitine (C3)	12.2	4.61 × 10^−5^
PC aa	PC aa C32:1	12.4	4.61 × 10^−5^
Biogenic Amines	Aminoadipic acid (alpha-AAA)	18.5	7.02 × 10^−4^
Aminoacids	Ornithine (Orn)	9.2	0.002
Acylcarnitines	Acetylcarnitine (C2)	9.1	0.003
SM	SM C18:1	8.3	0.006
Biogenic Amines	4-Hydroxyproline (t4-OH-Pro)	8.0	0.012
PC aa	PC aa C38:4	7.6	0.015
SM	SM C16:1	7.0	0.019
lysoPC	lysoPC a C16:1	7.3	0.021
Aminoacids	Lysine (Lys)	7.1	0.026
PC aa	PC aa C40:5	6.9	0.028
Acylcarnitines	Valerylcarnitine (C5)	6.5	0.037
**Negatively Associated**			
**Category**	**Metabolite**	**% Variation**	**FDR *p*-Value**
lysoPC	lysoPC a C18:2	−23.3	1.28 × 10^−22^
PC ae	PC ae C36:2	−20.5	3.11 × 10^−17^
PC ae	PC ae C34:3	−20.2	1.36 × 10^−16^
PC ae	PC ae C34:2	−18.2	2.18 × 10^−13^
PC ae	PC ae C40:6	−16.1	3.50 × 10^−10^
Aminoacids	Asparagine (Asn)	−16.1	5.24 × 10^−10^
PC ae	PC ae C40:1	−16.0	5.40 × 10^−10^
lysoPC	lysoPC a C18:1	−15.9	6.19 × 10^−10^
PC ae	PC ae C38:0	−15.2	1.59 × 10^−9^
lysoPC	lysoPC a C17:0	−14.3	7.32 × 10^−8^
Aminoacids	Glycine (Gly)	−13.6	1.98 × 10^−7^
PC aa	PC aa C38:6	−12.9	1.31 × 10^−6^
PC ae	PC ae C36:3	−12.3	4.71 × 10^−6^
PC aa	PC aa C38:0	−11.7	1.52 × 10^−5^
PC ae	PC ae C42:3	−11.9	1.52 × 10^−5^
Aminoacids	Histidine (His)	−11.8	1.61 × 10^−5^
PC aa	PC aa C36:0	−11.7	1.61 × 10^−5^
PC aa	PC aa C42:5	−11.6	3.00 × 10^−5^
PC ae	PC ae C36:1	−10.6	8.42 × 10^−5^
PC aa	PC aa C36:6	−10.6	9.45 × 10^−5^
PC ae	PC ae C30:0	−10.6	1.45 × 10^−4^
SM	SM C24:1	−10.5	1.51 × 10^−4^
PC ae	PC ae C40:5	−10.5	1.69 × 10^−4^
PC aa	PC aa C34:2	−10.5	1.91 × 10^−4^
PC ae	PC ae C42:2	−10.1	2.17 × 10^−4^
PC ae	PC ae C44:6	−10.3	2.32 × 10^−4^
Aminoacids	Serine (Ser)	−10.0	4.03 × 10^−4^
PC aa	PC aa C42:1	−10.0	4.78 × 10^−4^
PC ae	PC ae C34:1	−9.5	5.05 × 10^−4^
SM	SM C16:0	−9.7	5.06 × 10^−4^
PC aa	PC aa C42:6	−10.5	7.24 × 10^−4^
PC ae	PC ae C38:6	−9.4	7.72 × 10^−4^
PC ae	PC ae C32:1	−9.0	0.002
SM	SM C26:1	−8.9	0.002
SM	SM (OH) C22:2	−8.3	0.002
PC aa	PC aa C40:3	−8.8	0.002
PC ae	PC ae C42:1	−8.9	0.002
PC aa	PC aa C42:2	−9.6	0.003
PC aa	PC aa C42:0	−8.4	0.003
PC ae	PC ae C38:5	−8.5	0.003
PC ae	PC ae C42:4	−8.4	0.003
PC ae	PC ae C34:0	−7.8	0.006
PC ae	PC ae C36:5	−7.9	0.006
PC ae	PC ae C44:5	−7.8	0.008
PC aa	PC aa C40:2	−7.3	0.015
PC ae	PC ae C42:5	−7.2	0.015
PC ae	PC ae C36:0	−7.1	0.017
Biogenic Amines	Serotonin	−7.4	0.018
PC ae	PC ae C32:2	−6.7	0.019
PC ae	PC ae C38:4	−6.9	0.020
SM	SM (OH) C16:1	−6.7	0.021
Biogenic Amines	Creatinine	−6.1	0.023
Biogenic Amines	N-Acetylornithine (Ac-Orn)	−9.2	0.030
Aminoacids	Citrulline (Cit)	−6.2	0.030
Acylcarnitines	Dodecanoylcarnitine (C12)	−24.5	0.032
SM	SM (OH) C14:1	−5.9	0.041
PC ae	PC ae C38:2	−6.1	0.043

## Data Availability

The data presented in this study are available on request from the corresponding author.

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
