# Peer review of "Plasma Metabolomic Profiling in 1391 Subjects with Overweight and Obesity from the SPHERE Study"

_metabolites, 2021, doi:10.3390/metabo11040194_

Round 1

Reviewer 1 Report

Dear authors, 

I found your manuscript to be interesting, the experimental procedure well-designed and the results very well presented, with a satisfactory discussion section.

I believe a minor spell/grammar check is in order, and also maybe you could note in the limitations of your study that the majority of subjects were women and perhaps include/discuss previous literature reports (if available) regarding differences between men and women regarding plasma metabolomic profiles.

Reviewer 2 Report

The manuscript “Plasma metabolomic profiling in 1391 subjects with overweight and obesity from the SPHERE study” submitted by S. Fustinoni represents an original research article applying a targeted metabolomic effort to more than 1000 probands covering a broad range of BMI from slight overweight to pronounced obesity with the aim to characterize the plasma metabolom (188 metabolites) in both qualitative and quantitative terms. In fact, a number of positive and – more pronouncedly – negative correlations between certain metabolites and the BMI, among them certain species of lyso-phosphatidylcholine, phosphatidylcholine, aromatic, hydrophobic and branched-chain amino acids and hexose derivatives, have been identified. Some of which confirm, contradict or extend previous metabolomic studies conducted with similar patient cohorts.

The study is interesting and important, since focused on a relevant topic for the prediction and stratification of obesity and its complications including its preceding versions and based on the current golden state of methodology for biased metabolomics. The introduction represents a sound and straightforward argumentation of the unmet need left by previous studies. The results are well organized and accurately yet concisely describe the data obtained. The discussion gives a short summary of the results in critical comparison to previously published studies. Moreover, it offers mechanistic and physiological explanations for the changes observed, including pathway analysis, without too much of speculation and importantly and fairly presents the limitations of the study. The methods section provides the critical experimental parameters in sufficient detail under consideration of the relevant references.

I would like to ask the authors to address the following points:

  1. As the authors have correctly stated, there may be many confounding factors associated with obesity which are either not related to BMI at all or weaken or strengthen the relationship. Therefore it’s a pity that those factors, often but not always associated with BMI, such as insulin resistance, plasma insulin, glucose tolerance, blood glucose, have not been considered in the present analysis. Nevertheless, it would be very helpful for basic and clinical researchers in the areas of metabolic diseases to provide those data in an extended proband characterization (Table 1) as far as available.
  2. Recent findings about the release of glycosylphosphatidylinositol-anchored proteins together with lysoPC and PC from blood and tissue cells into the circulation of rats and humans suffering from obesity and diabetes (Muller et al. 2019, 2020) may shed some light on the negative correlation between BMI and the majority of plasma lysoPC and PC species. They found that the released GPI-anchored proteins are preferably degraded by a specific serum phospholipase D leading to decreased plasma levels of those proteins compared to control probands. This type of counterregulation operating in the obese/diabetic state could also apply to lysoPC and PC, associated together with the GPI-anchored proteins in micelle-like complexes, causing lowering of their plasma levels. Muler et al. suggested upregulation of lipolytic degradation as a protective mechanism against the deleterious effects of amphiphilic GPI-anchored proteins (as might hold true for lysoPC?, PC?) at elevated levels in plasma.
  3. Why did the authors restrict pathway analysis (Figure 6) to alterations in amino acid concentrations and excluded the lipid changes?

Minor point:

line 296: something is wrong with this sentence “and very a lot decreased”

Reviewer 3 Report

The paper is interesting as it looks at the association of BMI with the metabolomic profile. The following would improve the presentation of the paper:

  1. A method section to summarize all the methods used can be added.
  2. Please provide a reason why the Tobit linear regression is used. I assume metabolites can be measured within certain levels. Please explain whether the data was left or right-censored.
  3. Please explain whether the metabolites data was standardized prior to fitting the Tobit models.
  4. Was BMI considered as a continuous or categorical variable for inclusion in the Tobit models? Would results differ if BMI is included as a continuous or categorical variable?
  5. How were the levels of 0.4 and 0.7 for statistically significant correlations selected in Figures 3 and 4?
  6. Is ethnicity expected to play a role in the association between BMI and the metabolomic profile? Is ethnicity available for the current cohort? If yes, can it be shown in Table 1?
  7. Are there other details available on the occupation type, such as sedentary or physically active workers?
